# Simulation Uncertainty for a Virtual Ultrasonic Flow Meter

**Martin Straka** [1,*], **Andreas Weissenbrunner** [1], **Christian Koglin** [1], **Christian Höhne** [2] **and Sonja Schmelter** [1]

1 Physikalisch-Technische Bundesanstalt (PTB), Abbestraße 2-12, 10587 Berlin, Germany;
andreas.weissenbrunner@ptb.de (A.W.); christian.koglin@ptb.de (C.K.); sonja.schmelter@ptb.de (S.S.)
2 FLEXIM Flexible Industriemesstechnik GmbH, Boxberger Straße 4, 12681 Berlin, Germany;
choehne@flexim.de
\* Correspondence: martin.straka@ptb.de

**Abstract:** Ultrasonic clamp-on meters have become an established technology for non-invasive flow measurements. Under disturbed flow conditions, their measurement values must be adjusted with corresponding fluid mechanical calibration factors. Due to the variety of flow disturbances and installation positions, the experimental determination of these factors often needs to be complemented by computational fluid dynamics (CFD) simulations. From a metrological perspective, substituting experiments with simulation results raises the question of how confidence in a so-called virtual measurement can be ensured. While there are well-established methods to estimate errors in CFD predictions in general, strategies to meet metrological requirements for CFD-based virtual meters have yet to be developed. In this paper, a framework for assessing the overall uncertainty of a virtual flow meter is proposed. In analogy to the evaluation of measurement uncertainty, the approach is based on the utilization of an expanded simulation uncertainty representing the entirety of the computational domain. The study was conducted using the example of an ultrasonic clamp-on meter downstream of a double bend out-of-plane. Nevertheless, the proposed method applies to other flow disturbances and different types of virtual meters. The comparison between laboratory experiments and simulation results with different turbulence modeling approaches demonstrates a clear superiority of hybrid RANS-LES models over the industry standard RANS. With an expanded simulation uncertainty of $1.44 \times 10^{-2}$, the virtual measurement obtained with a hybrid model allows for a continuous determination of calibration factors applicable to the relevant mounting positions of a real meter at a satisfactory level of confidence.

**Keywords:** virtual measurement; simulation uncertainty; digital metrological twin; validation; ultrasonic flow meter; clamp-on

## 1. Introduction

Ultrasonic clamp-on meters have become an established technology for non-invasive flow measurements in industrial applications. As a portable device, they can be used at multiple measurement positions without interrupting the ongoing processes and are suitable for a wide range of fluids, temperatures, and conduit sizes. On the downside, the comparatively simple measuring principle can result in large measurement deviations under non-ideal (disturbed) flow conditions caused by bend configurations, valves, reductions, pumps, and other components installed upstream. For this reason, the measurement value has to be adjusted by means of fluid mechanical calibration factors, hereinafter referred to as $K_d$. In that regard, $K_d$ acts as a constant of proportionality between the mean velocity in the pipe and the averaged velocity over the ultrasonic path as determined by the meter. Its magnitude can vary significantly ($\approx \pm 15\%$) depending on the shape of the velocity distribution (flow conditions) in the pipe. Similar to other flow properties, specific values of $K_d$ are usually stored in the meter's processing unit.

Prior to implementation, $K_d$ can be determined experimentally on flow meter test rigs. However, for each single configuration, i.e., for each flow disturbance, downstream

position, and angle of installation, a separate measurement is needed. Since this makes a fully experimental approach unfeasible, measurements on test rigs are often complemented by simulation results. When a real measurement process is replaced by a simulation, this can be referred to as a *virtual measurement* or *virtual meter*. Modeling a virtual flow meter can be achieved by means of computational fluid dynamics (CFD). In contrast to an individual experiment, the virtual measurement realized in a single CFD simulation generates approximations of $K_d$ for all possible angular orientations and downstream positions. Substituting laboratory experiments with virtual measurements allows the examination of a great number of disturbances, but it also raises the issue of how confidence in simulation results can be ensured.

In metrology, confidence in measurement results is established through a quantified statement of the associated measurement uncertainty. For the most part, it is reported in the form of an expanded uncertainty to provide a suitable confidence interval. The current standard for the calculation of uncertainties is the *guide to the expression of uncertainty in measurement* (GUM) [1] and its supplements [2,3]. By contrast, metrological standards for virtual measurements do not yet exist. In an attempt to summarize some important requirements for virtual meters, Eichstädt et al. [4] introduced the concept of a *digital metrological twin*. As stated in Poroskun et al. [5], this term was explicitly introduced to clarify its affiliation with metrology, whereas the definition of the term *digital twin* may differ significantly for other applications, see, e.g., [6–8]. Two essential requirements of the *digital metrological twin* applicable to a virtual flow meter are that uncertainties are calculated according to valid standards and that it is validated by traceable measurements.

Due to its diverse application range, there are several approaches for quantifying errors and uncertainties in CFD simulations. According to Le Maître and Knio [9], simulation errors can be categorized into data, numerical, and modeling errors. Data errors arise from incorrect specifications of the boundary conditions, fluid properties, or geometrical dimensions. Corresponding uncertainties can be quantified using multiple simulations with different input parameters, see, e.g., [10–12]. Yet in a laboratory environment, they are expected to be small compared to other error types. Numerical errors correspond to the difference between an analytical solution and its approximations resulting from the discretization of the underlying equations. This includes, e.g., the mesh resolution, time step size, or discretization schemes. The elimination of numerical errors or quantification of corresponding uncertainties is typically addressed within the process of verification. Modeling errors are related to the formulation of the mathematical equations approximating the exact physics. In CFD simulations, these errors are usually connected to a simplified representation of turbulent flow phenomena. Their magnitudes can vary significantly depending on the chosen turbulence model. Since modeling errors are generally difficult to assess, they are normally determined through a comparison with experimental data during the process of validation. Verification and validation are well-known and recognized procedures to assure quality in CFD predictions, see, e.g., [13,14]. While they are usually allocated to numerical or modeling errors only and, thus, carried out successively, studies such as [15,16] have shown that the sources of errors cannot always be separated. In any case, an uncertainty assessment for a virtual meter in conformity with metrological standards requires not only the evaluation of different uncertainty contributions but also a strategy as to how they can be combined and utilized.

In this paper, we propose a framework for assessing the overall simulation uncertainty of a CFD-based virtual flow meter. In analogy to the evaluation of measurement uncertainty according to the GUM [1], the approach builds upon the utilization of an expanded simulation uncertainty that provides a confidence interval of 95.45%. Quantifying the uncertainty is realized through a validation experiment. The study was conducted using the example of calibration factors for an ultrasonic clamp-on meter installed downstream of a double bend out-of-plane. Since all installation positions are equally relevant in practical applications, the simulation uncertainty is intended to represent the entirety of the computational domain. In the remainder of the paper, the experimental determination of

$K_d$ and quantification of the associated measurement uncertainty are covered in Section 2, while the simulation-based approximation of $K_d$ is addressed in Section 3. Apart from a description and verification of the simulation setup, the latter includes a visualization of the flow patterns and a comparison with the experimental results. The derivation of the expanded simulation uncertainty is presented in Section 4.

## 2. Experimental Determination of Calibration Factors

This section covers the experimental part of the investigations and includes a description of the test facility as well as the installation of the ultrasonic flow meters downstream of a double bend out-of-plane. Following an explanation of the measuring principle, the methodological approach leading to the experimental determination of the fluid mechanical calibration factors $K_d$ is described. A model equation for the combined measurement uncertainty is derived. Furthermore, the expanded measurement uncertainty is calculated in accordance with the GUM [1].

### 2.1. Measurement Setup

Clamp-on measurements were performed at the long-term ultrasonic and laser measurement facility (LULA) at the national metrology institute of Germany (PTB). A schematic representation of the test rig, the double bend out-of-plane, and the installation of the ultrasonic meters are depicted in Figure 1.

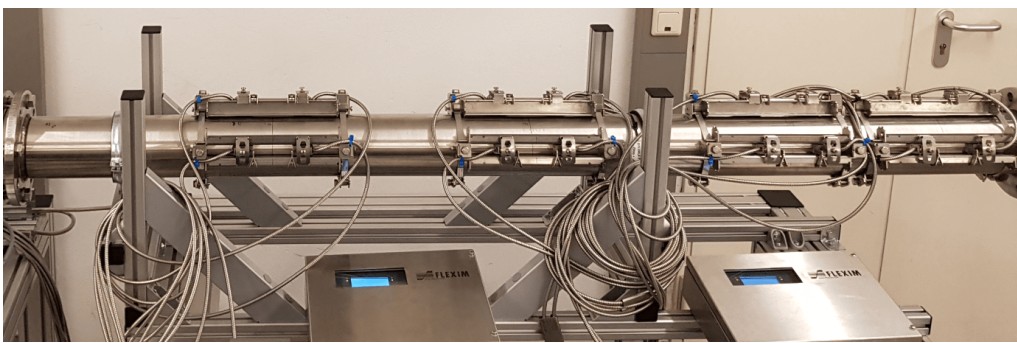

(**a**) Ultrasonic clamp-on meters (FLEXIM FLUXUS F721) mounted onto the 2 m pipe section

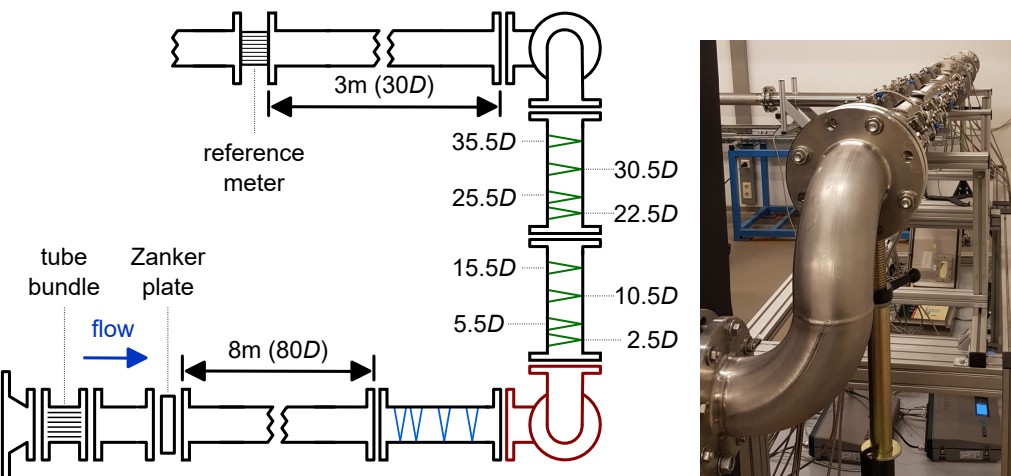

(**b**) Schematic representation of the test rig. (**c**) Double bend out-of-plane.

**Figure 1.** Measurement setup at the long-term ultrasonic and laser measurement facility (LULA). In (**b**), the blue and green V-shaped markers represent the positions of the ultrasonic meters installed upstream and downstream of the double bend, respectively.

### 2.1.1. Flow Conditions

Measurements were taken at a water temperature of $30\,°C$ (kinematic viscosity $\nu = 0.8 \times 10^{-6}\,\mathrm{m^2\,s^{-1}}$) and a flow rate of $\dot{V} = 11.32\,\mathrm{m^3\,h^{-1}}$. This corresponds to a volumetric velocity of $v_{vol} = \dot{V}/A = 0.4\,\mathrm{m\,s^{-1}}$ (where $A$ is the pipe cross-section) and a friction velocity of $u_\tau = 2.06 \times 10^{-2}\,\mathrm{m\,s^{-1}}$, a Reynolds number of $Re = v_{vol} \cdot D/\nu = 5 \times 10^4$ , and a friction Reynolds number of $Re_\tau = u_\tau \cdot D/\nu = 1288$.

### 2.1.2. Test Rig

In the configuration illustrated in Figure 1b, LULA provides measurement sections for the investigation of fully developed flow conditions and disturbed flow conditions downstream of a double bend out-of-plane. The flow rate control was realized by means of an electromagnetic reference meter (KROHNE WATERFLUX 3070) that was calibrated at the gravimetric heat meter calibration facility at the PTB. Its combined relative uncertainty and degrees of freedom ($dof_i$) at a Reynolds number of $5 \times 10^4$ are given by $u_r(\dot{V}_{ref}) = 7.74 \times 10^{-4}$ and $dof_i = 83$ (Straka [17]). All pipes and bends were constructed in the nominal diameter DN 100 with an internal diameter ($D$) of 100.0 mm ($\pm 50\,\mu m$) and a wall roughness $R_a \approx 5\,\mu m$, which corresponded to a dimensionless roughness parameter $ks^+ \approx 0.7$. It can therefore be considered as hydraulically smooth; compare to e.g., Gersten [18]. To minimize the flange offset, they were connected by centering rings with a sliding fit tolerance of $40\,\mu m$. The double bend depicted in Figure 1c consisted of two identical welded elbow fittings with a curvature radius ($R_c$) of 142.5 mm ($R_c/D = 1.425$). The measurement section downstream had a length of 4 m, which corresponded to $40\,D$. To ensure fully developed flow conditions, the measurement section upstream of the double bend was preceded by a flow conditioning unit, which combined a pipe reducer (159 mm to 100 mm) with a tube bundle straightener and a Zanker plate according to ISO 5167-1 [19], see Straka et al. [20]. For investigations of fully developed flow conditions, the free inlet length added up to 8 m ($80\,D$), see Figure 1b. Similarly, the double bend was preceded by a straight inlet section of 10 m ($100\,D$).

### 2.1.3. Clamp-On Meters

Throughout the measurement campaign, a total of eight clamp-on meters (FLEXIM FLUXUS F721) were permanently mounted onto a 2 m pipe section, see Figure 1a. The meters were positioned at four locations along the pipe axis ($z$) and two angular orientations ($\varphi$) at 18° and 108°. For the determination of the calibration factors, this unit was installed both in the first and second half of the measurement section downstream of the double bend, resulting in $N_z = 8$ downstream locations with normalized distances ($z/D$) of 2.5, 5.5, 10.5, 15.5, 22.5, 25.5, 30.5 and 35.5. In each half, the pipe was rotated around the pipe axis by 45°, resulting in additional angular orientations of 63° and 153°. With $N_\varphi = 4$, the total number of measurement positions was given by $N_z \cdot N_\varphi = 32$.

### 2.2. Ultrasonic Flow Rate Measurements

As illustrated in Figure 2a, the ultrasonic clamp-on meter measured a transit time difference ($\Delta t$) between the upstream and downstream signals transmitted and received by a pair of transducers $A$ and $B$. Using the delay time $t_0$, the transit time at zero flow $t_{tr}$ and the path-geometry factor $K_g$, a path velocity proportional to $\Delta t$ was determined by

$$v_{path} = K_g \cdot \frac{\Delta t}{2\,(t_{tr} - t_0)} \ . \tag{1}$$

As a constant of proportionality, $K_g$ has the unit of velocity. In general, volume flow rates ($\dot{V}$) are determined as $\dot{V} = A \cdot v_{vol}$. Applied to the ultrasonic flow rate measurement using $v_{path}$, $\dot{V}$ is calculated as

$$\dot{V} = A \cdot \underbrace{K_u \cdot K_d}_{K_p} \cdot v_{path} \, . \tag{2}$$

In the equation, $K_p$ represents the combined fluid mechanical calibration factor that accounts for the fact that $v_{path}$ does not comply with the volumetric velocity $v_{vol}$. For the implementation within the meter, it is useful to split $K_p$ into $K_u$ and $K_d$, representing the correction for a fully developed (undisturbed) turbulent flow and a particular flow disturbance, respectively. Values for $K_u$ can be determined experimentally or estimated using a theoretical velocity profile, e.g., the semi-analytical approach from Gersten and Herwig [21]. For the ultrasonic clamp-on meter in the reflection mode, the theoretical range of $K_u$ is from 0.75 for the laminar flows to 1 for the infinitely large Reynolds numbers in a perfectly smooth pipe. In practical applications, $K_u$ typically ranges between 0.88 and 0.98 in the turbulent regime. $K_d$, on the other hand, can typically range between 0.8 and 1.2, mainly depending on the particular flow disturbance, its downstream distance, and angular orientation towards the meter.

For the current setup, the expanded relative uncertainty of the clamp-on meters with respect to the determination of $\dot{V}$ under fully developed flow conditions is roughly estimated as $U_r(\dot{V}) \approx 1\%$ ($k = 2$); see Appendix A.1.

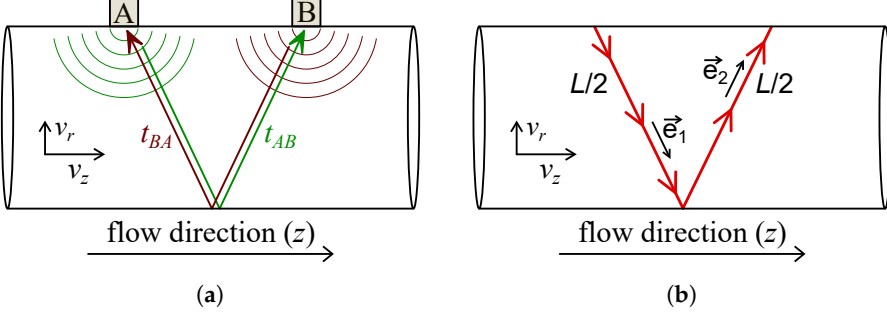

(**a**)            (**b**)

**Figure 2.** (**a**) Schematic illustration of an ultrasonic flow meter in the reflection mode using the transit time difference method. Source: Straka et al. [22] (CC BY 4.0). (**b**) Implementation of the measuring principle in the velocity field provided by the CFD simulation.

### 2.3. Derivation of the Calibration Factors

When $K_d$ is determined experimentally, this approximation is hereinafter referred to as $K_{d.E}$. Device independent values of $K_{d.E}$ were obtained by comparing the results from two separate experiments. At first, the clamp-on flow meters were exposed to fully developed flow conditions, see Figure 1b. In the second experiment, the flow meters were installed downstream of the double bend. In each measurement, the combined calibration factor $K_p$ was determined by comparing the flow rate provided by the ultrasonic flow meter ($\dot{V}$) and the reference meter of the test rig ($\dot{V}_{ref}$) according to

$$K_p = \frac{\dot{V}_{ref}}{\dot{V}} \, . \tag{3}$$

From the first measurement series, $K_u$ can directly be determined from Equation (3) because, under fully developed conditions, $K_d$ equals one and, thus, $K_p = K_u$. From the second measurement series, $K_{d.E}$ can then be extracted by first calculating $K_p$ according to Equation (3) under disturbed flow conditions and then dividing it by $K_u$ according to

$$K_{d.E} = \frac{K_p}{K_u} \, . \tag{4}$$

At every measurement position, $K_p$ is calculated as the arithmetic means of three consecutive measurements, each taken within a time interval of 20 min. Note that most of the input quantities for $\dot{V}$ included in Equation (2) remain constant during the measurement campaign. As a result, they can be neglected in the uncertainty analysis for $K_{d.E}$. As opposed to an individual uncertainty evaluation for $K_p$ or $K_u$, the impact of $u_r(\dot{V})$ is thus significantly reduced, see Appendix A.2.

*2.4. Measurement Uncertainty*

The uncertainty associated with the experimental determination of the calibration factors $K_{d.E}$ is calculated in accordance with the GUM [1]. In Section 4, it will be used for the validation of the virtual measurements and evaluation of the simulation uncertainty.

2.4.1. Combined Uncertainty $u_c(K_{d.E})$

In its primary form, the model equation for the combined relative measurement uncertainty $u_{c,r}(K_{d.E})$ derived from Equation (4) is given by

$$u_{c,r}^2(K_{d.E}) = u_r^2(K_u) + u_r^2(K_p).$$

(5)

For the uncertainty evaluation, the terms on the right side of Equation (5) must be divided into manageable components. A detailed description of the individual uncertainty contributions is included in Appendix A.2. For the validation of the simulation results in Section 4, the combined (absolute) measurement uncertainty $u_c(K_{d.E}) = u_{c,r}(K_{d.E}) \cdot K_{d.E}$ is used. Throughout the measurement positions, $u_c(K_{d.E})$ ranges from $1.50 \times 10^{-3}$ to $3.33 \times 10^{-3}$. These variations are largely due to the strongly location-dependent uncertainty components associated with the angular and downstream alignment as well as the reproducibility of the individual sensors, see Appendix A.2.

2.4.2. Expanded Uncertainty $U(K_{d.E})$

As usual, in metrology, uncertainties are multiplied by a coverage factor $k \approx 2$ to obtain a confidence interval of 95.45%. This results in the expanded measurement uncertainty given by $U(K_{d.E}) = k \cdot u_c(K_{d.E})$ and requires the specification of the effective degrees of freedom ($\mathrm{dof}_{eff}$). As recommended in the GUM [1], $\mathrm{dof}_{eff}$ is calculated by using the Welch—Satterthwaite equation, given by

$$\mathrm{dof}_{eff} = u_c^2 \cdot \left[ \sum_{i=1}^{N} \frac{u_i^4}{\mathrm{dof}_i} \right]^{-1},$$

(6)

where $u_i$ and $\mathrm{dof}_i$ represent the individual uncertainty components of $u_c(K_{d.E})$ and their associated degrees of freedom. Using $\mathrm{dof}_{eff}$ and the required confidence interval of 95.45%, $k$ can be obtained from the inverse function of the Student's t-distribution. A representative calculation example for $z/D = 35.5$ and $\varphi = 18°$ is presented in Appendix A.2, Table A1. Throughout the measurement positions, $U(K_{d.E})$ ranges from $3.05 \times 10^{-3}$ to $6.66 \times 10^{-3}$ ($k \approx 2$). All values are depicted in Figure 3b.

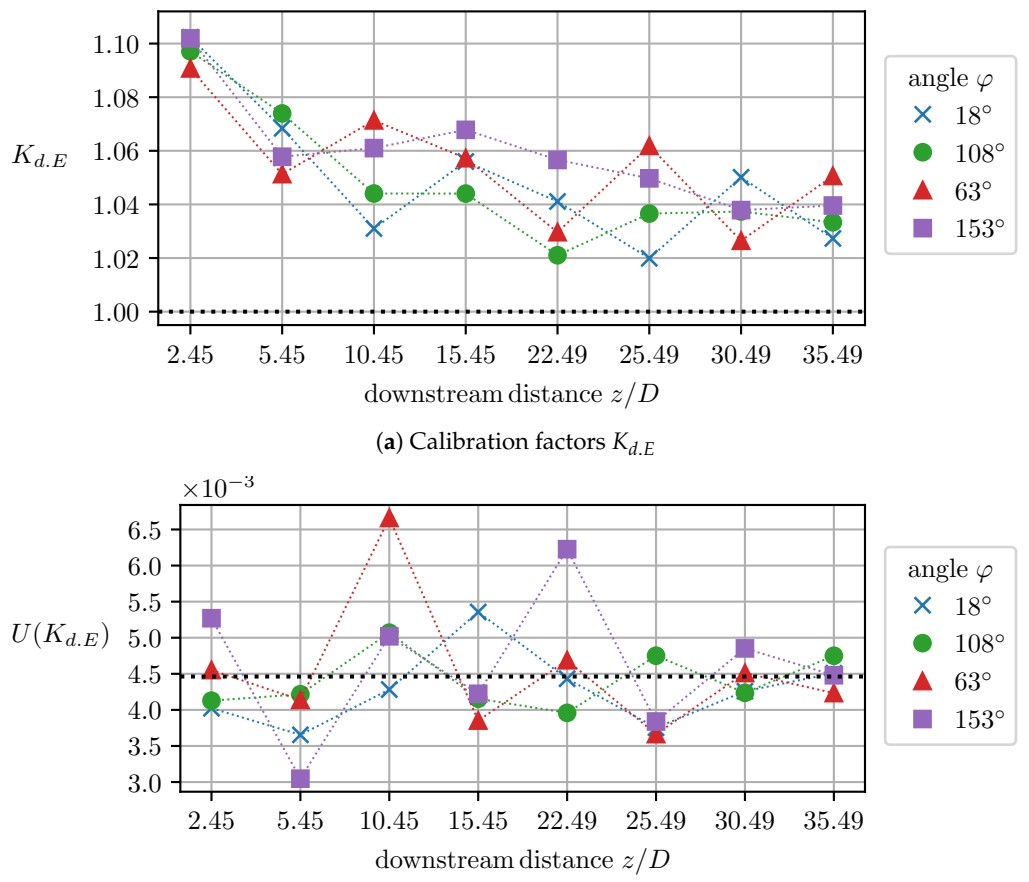

(**a**) Calibration factors $K_{d.E}$

(**b**) Expanded measurement uncertainty $U(K_{d.E})$ ($k \approx 2$)

**Figure 3.** Ultrasonic clamp-on measurement results downstream of the double bend out-of-plane.

## 2.5. Experimental Results

The experimental results of $K_{d.E}$ at the 32 measurement positions are depicted in Figure 3a. It can be noticed that the magnitude of $K_{d.E}$ tends to decrease with growing distance to the double bend out-of-plane, although the value corresponding to fully developed flow conditions ($K_{d.E} = 1$) is clearly not reached at $z/D = 35.5$. Furthermore, a strong angular dependency can be detected. Thus, an interpolation between the measurement positions does not appear to be meaningful.

## 3. Simulation-Based Determination of Calibration Factors

The current section deals with the realization of the virtual ultrasonic clamp-on measurements. This includes a description of the simulation setup and the implementation of the measuring principle that allows a simulation-based determination of $K_d$. In analogy to the experimental terminology, the approximation of $K_d$ obtained by means of a virtual meter is referred to as $K_{d.S}$. Since the selection of the turbulence model represents the largest source of modeling errors, a variety of different turbulence models are investigated. Using today's state-of-the-art Reynolds-averaged Navier-Stokes (RANS) models often results in poor predictions of complex flows, while scale-resolving approaches, such as large eddy simulations (LES), remain unfeasible for industrial applications. In between, hybrid RANS-LES models represent a means of combining comparatively low modeling errors and reasonable computing times. Simulation results are presented for the Spalart–Allmaras RANS and hybrid SBES-SST models. Apart from an examination of $K_{d.S}$ throughout the computational domain, an analysis of both axial and secondary velocity profiles is presented. Three different numerical uncertainties and the uncertainty due to the time averaging intervals are evaluated within a series of verification studies. Finally, the measurement and simulation results, $K_{d.S}$ and $K_{d.E}$, are compared.

### 3.1. Simulation Setup

Simulations were performed with the commercial software package ANSYS Fluent 2020 R1 using the pressure-based solver. Meshes for the double bend geometry and adjacent pipe sections were created with ANSYS ICEM CFD 2020 R1.

### 3.1.1. Turbulence Modeling

Steady-state simulations were performed with a variety of different RANS models. These included the standard $k$-$\epsilon$ (Launder and Spalding [23]), realizable $k$-$\epsilon$ (Shih et al. [24]), standard $k$-$\omega$ (Wilcox [25]), and SST $k$-$\omega$ (Menter [26]) models from the group of two-equation eddy-viscosity models and the one-equation Spalart–Allmaras model (Spalart and Allmaras [27]). Furthermore, three different Reynolds stress models were tested. Convergence could be achieved with the linear pressure–strain (Launder and Shima [28]) and Stress-$\omega$ (Wilcox [25]) models, whereas the quadratic pressure–strain model (Speziale et al. [29]) failed in this regard. Transient simulations with hybrid RANS-LES models were carried out with the stress-blended eddy simulation (SBES) model (Menter [30]) and the detached eddy simulation (DES) model (Spalart et al. [31]). Each was set up with both the realizable $k$-$\epsilon$ and the SST $k$-$\omega$ RANS models. Low-Reynolds corrections were enabled for the $k$-$\omega$-based models. Apart from that, all RANS models were used in accordance with their default settings and constants in ANSYS Fluent 2020 R1. For SBES, the location-dependent blending function specified in Straka et al. [32] with a modified constant of $a = 100$ as well as the Smagorinsky–Lilly sub-grid model (Smagorinsky [33]) with $C_S = 0.025$ was used.

### 3.1.2. Solver Settings

Transient simulations were run with a time step size of 7.5 ms. Following an initial adaptation period of 8000 time steps (60 s), the instantaneous velocity fields were averaged over 16,000 time steps (120 s). Further solution methods are listed in Table 1.

**Table 1.** Applied solution methods and residuals in ANSYS Fluent 2020 R1.

|  | **Hybrid RANS-LES** | **RANS** |
| --- | --- | --- |
| **Pressure-Velocity Coupling** | **SIMPLE** | **Coupled** |
| Discretization schemes | | |
| Temporal | Bounded Second-order Impl. | — |
| Spatial | | |
|   Diffusion | Central differencing | Central differencing |
|   Momentum | Central differencing | Second-order upwind |
|   Pressure | Second-order | Second-order |
|   Gradient | Green-Gauss cell-based | Least-Squares cell-based |
|   Turbulent quantities | First-order upwind | Second-order upwind |
| Scaled residuals | | |
|   Continuity, velocities | $1 \times 10^{-5}$ | $1 \times 10^{-15}$ |
|   Turbulent quantities | $1 \times 10^{-4}$ | $1 \times 10^{-15}$ |

### 3.1.3. Geometry and Meshing

As illustrated in Figure 4a, the computational domain consists of a straight pipe section of 1.0 m (10 $D$), a double bend, and a subsequent pipe section of 6.4 m (64 $D$). The hexahedral mesh has an O-grid structure depicted in Figure 4b, which is swept along the pipe axis. Two mesh versions were used for the hybrid RANS-LES and RANS simulations. With 136 and 45 cells in the circumferential and radial direction, a non-dimensional wall distance of $y^+ \approx 1.0$, and 43 cells per diameter in the streamwise direction, the mesh versions for the hybrid simulations have $\approx 1.64 \times 10^7$ cells. For RANS, the streamwise grid resolution in the straight pipe sections decreased towards the inlet and outlet from 43 to 17 cells per diameter using an exponential growth rate, resulting in a reduced mesh size of $\approx 9.50 \times 10^6$ cells.

### 3.1.4. Boundary Conditions

For all simulations, a fully developed velocity distribution and turbulence quantities from a precursor RANS simulation with either the SST-*k*-$\omega$ or standard *k*-$\epsilon$ model were utilized at the inlet boundary. In the transient simulations, the vortex method (Mathey et al. [34]) with 250 vortex points was applied to create velocity fluctuations. Furthermore, a no-slip condition and the zero diffusion flux outflow condition were used at the pipe wall and outlet boundaries, respectively.

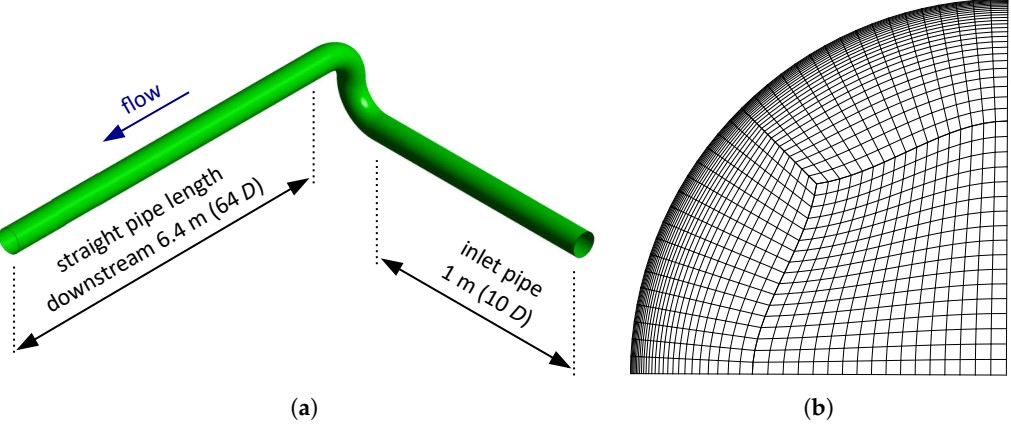

(a)                                                     (b)

**Figure 4.** (**a**) Double bend out-of-plane (curvature radius $R_c/D = 1.425$) and adjacent pipe geometry. The length of the downstream distance is not shown to scale. (**b**) One-quarter of the multi-block-structured O-grid in the pipe cross-section.

### 3.2. Implementation of the Measuring Principle

Similar to the experimental determination of $K_{d.E}$, $K_{d.S}$ was derived by means of the path velocity $v_{path}$. Yet, in contrast to the transit time measurement of the clamp-on meters, $v_{path}$ was calculated from the simulation results as the average velocity along the ultrasonic path geometry in the velocity field $\vec{v}$, according to

$$v_{path} = \frac{1}{Le_z} \left( \int_0^{L/2} \vec{v} \cdot \vec{e_1} \, \mathrm{d}l + \int_0^{L/2} \vec{v} \cdot \vec{e_2} \, \mathrm{d}l \right), \tag{7}$$

where $L$ is the length of the V-shaped ultrasonic path and $\vec{e_1}$, $\vec{e_2}$ are the unit vectors of the two path sections, see Figure 2b. $K_{d.S}$ can be calculated by comparing the path velocities of an undisturbed reference profile and the disturbed velocity profiles according to

$$K_{d.S} = \frac{v_{path.u}}{v_{path.d}} . \tag{8}$$

Evaluating the fully developed profile obtained from a separate simulation of a straight pipe using the SBES-SST model yields $v_{path.u} = 1.082 \cdot v_{vol}$. Since this value varied only slightly for the different turbulence models, it was used for all simulations.

### 3.3. Simulation Results

Results are presented for the Spalart–Allmaras and SBES-SST turbulence models. Since the simulations provide the flow field in the whole 3D domain, $K_{d.S}$ was accessible at any angular orientation and downstream distance. Flow profiles were examined to explain the specific characteristics in the measurement and simulation results.

### 3.3.1. Spatial Distribution

Figure 5 shows that $K_{d.S}$ propagates throughout the domain in a wave-like manner. Its frequencies and amplitudes continuously diminished with increasing distance to the double bend. Yet at the maximum distance of $64\,D$, the flow disturbance was not fully subsided. Furthermore, the wavefronts did not run parallel to the vertical axis. This explains the

angular dependency of the measurement data observed in Figure 3. While the qualitative development of $K_{d.S}$ was predicted similarly by both models, the specific arrangement of the wave crests, as well as their phases and magnitudes, were different.

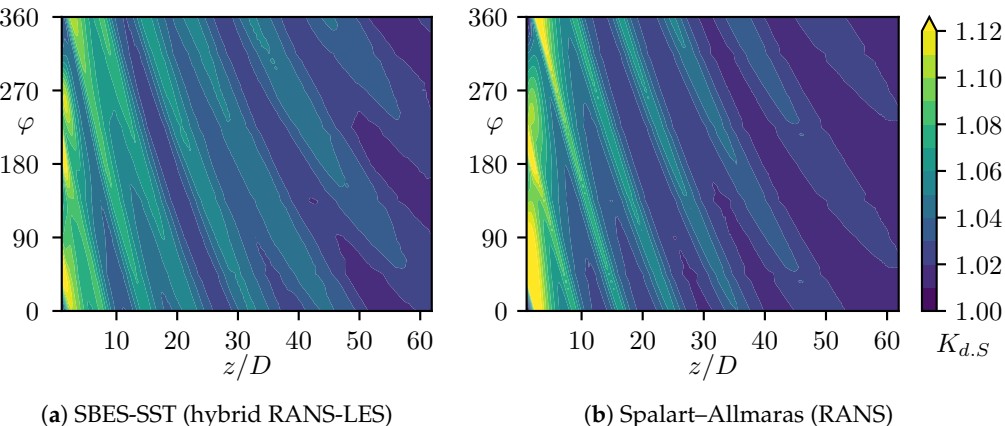

(**a**) SBES-SST (hybrid RANS-LES)     (**b**) Spalart–Allmaras (RANS)

**Figure 5.** Calibration factor $K_{d.S}$ calculated from the simulation results acc. Equation (8) as a function of the angular orientation ($\varphi$) and downstream distance ($z/D$).

### 3.3.2. Velocity Profiles

A velocity profile is defined as the velocity distribution over the pipe cross-section at a particular downstream position. Fully developed pipe flows are characterized by a symmetrical distribution of the time-averaged axial profile and the absence of secondary components. Velocity profiles downstream of the double bend, as predicted by the SBES-SST and Spalart–Allmaras model, are depicted in Figure 6. Throughout the domain, the axial profiles show a radial displacement of the core velocity region towards the pipe wall. In general, asymmetry can be connected to the angle-dependent fluctuations observed for $K_d$, as the average velocity along the individual ultrasonic paths ($v_{path}$) varies accordingly. In comparison to fully developed conditions, the axial profiles have overall flatter shapes. As a result, $v_{path}$ takes on comparatively smaller values and $K_d$ is thus consistently larger than 1. In addition to the deformation, the axial velocity profiles describe a helical movement around the pipe axis in the clockwise direction. This correlates with the single vortex structure (swirl) present in the secondary flow, see Figure 6b. The rotation is the reason for the wave-like patterns of $K_{d.S}$ in Figure 5, i.e., a rotation of 360° corresponds to the length between two adjacent wave crests.

Comparing the two turbulence models, it is noticeable that the shapes of the axial profiles, especially in the near-field range up $z/D = 5$, are qualitatively different. At $z/D = 60$, the progression towards the fully developed state is much further advanced in the case of the RANS model. This is consistent with the comparatively lower values of $K_{d.S}$ in the far-field range observed in Figure 5. As for the secondary velocity profiles in Figure 6b, the swirling velocity is slightly higher in the case of the SBES-SST model. As a result, there is a noticeable phase shift between the axial profiles in Figure 6a.

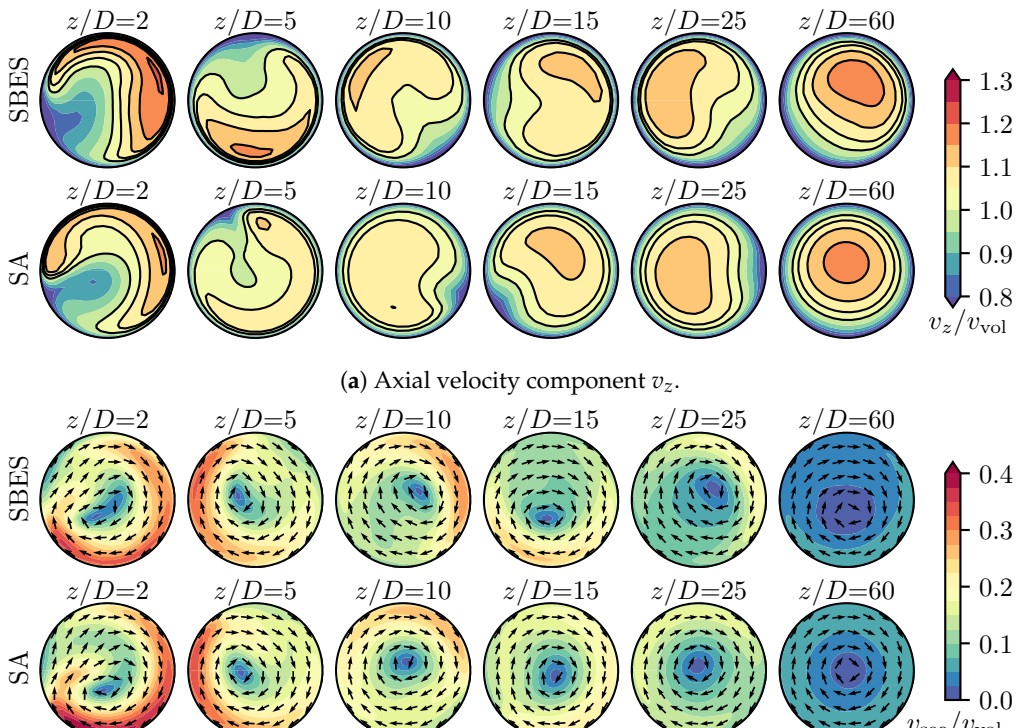

(**a**) Axial velocity component $v_z$.

(**b**) Amplitude and direction of secondary velocity components $v_{\text{sec}} = (v_x, v_y)^T$.

**Figure 6.** Velocity profiles downstream of the double bend out-of-plane.

### 3.4. Calculation Verification

Numerical uncertainties and the uncertainties due to the time-averaging interval were evaluated as part of a calculation verification. All contributions were combined in the calculation uncertainty $u_{cal}(K_{d.S})$, which will be used for the comparison between $K_{d.E}$ and $K_{d.S}$ in Section 3.5.

#### 3.4.1. Numerical Uncertainties

Three independent studies covering the overall spatial discretization ($u_{disc}$), wall-normal grid resolution at the wall ($u_{wall}$), and iterative convergence within time steps ($u_{iter}$) were conducted. In the case of the transient simulations, $u_{disc}$ also included temporal discretization. As for RANS, $u_{iter}$ was neglected, given that scaled residuals of $1 \times 10^{-15}$ were used. In each study, simulations with three different refinement levels were evaluated. Their parameters are listed in Table 2.

**Table 2.** Variable parameters in the verification studies regarding the numerical uncertainties.

| Uncertainty | | Variable Parameter (s) | Refinement Level | | |
|---|---|---|---|---|---|
| Error Source | Symbol | | 1 | 2 | 3 |
| spatial & temporal discretization | $u_{disc}$ | cells on circumference | 120 | 136 | 160 |
| | | streamwise cells/diameter | 35 | 43 | 50 |
| | | cells in radial direction | 41 | 45 | 51 |
| | | time step size [ms] | 10 | 8 | 7 |
| spatial discretization wall-normal direction | $u_{wall}$ | wall-normal distance $y^+$ | 1.0 | 0.5 | 0.2 |
| iterative convergence within time steps | $u_{iter}$ | scaled momentum residual | $10^{-5}$ | $10^{-6}$ | $10^{-7}$ |

The Richardson extrapolation is a common method used to calculate and correct numerical errors, see, e.g., Roache [16]. This approach requires an asymptotic convergence for a parameter, such as the grid size when the level of refinement is increased. In case of oscillatory convergence, Richardson extrapolation fails, and errors cannot be derived or corrected. As an alternative, Stern et al. [14] suggest stating numerical uncertainties as one-half of the difference between the highest and lowest values of the acquired target variable. Given that $K_{d.S,\max}$ and $K_{d.S,\min}$ represent the maximum and minimum within a verification study, the local numerical uncertainty component is thus calculated as

$$u_{num,local} = \frac{1}{2} \cdot \left( K_{d.S,\max} - K_{d.S,\min} \right). \tag{9}$$

A representation for the entire domain is obtained by calculating the arithmetic mean over $\varphi \in [0, 2\pi)$ and $z/D \in [2, 20]$ in steps of $\pi/60$ and 1, respectively. In all studies for both models, the different solutions show an oscillatory convergence. Thus, $u_{disc}$, $u_{wall}$, and $u_{iter}$ are estimated in accordance with Equation (9). Results are listed in Table 3.

**Table 3.** Calculation uncertainties associated with the simulation-based determination of $K_d$.

| Uncertainty | | Turbulence Model | |
|---|---|---|---|
| **Related to** | **Symbol** | **SBES-SST (Hybrid)** | **Spalart–Allmaras (RANS)** |
| Time-averaging | $u_{time}$ | $1.32 \times 10^{-3}$ | – |
| Grid & time step size | $u_{disc}$ | $1.65 \times 10^{-3}$ | $2.73 \times 10^{-3}$ |
| Wall-normal distance | $u_{wall}$ | $1.52 \times 10^{-3}$ | $1.14 \times 10^{-3}$ |
| Iterative convergence | $u_{iter}$ | $1.13 \times 10^{-3}$ | – |
| Calculation | $u_{cal}(K_{d.S})$ | $2.84 \times 10^{-3}$ | $2.96 \times 10^{-3}$ |
| | $U_{cal}(K_{d.S})$ | $5.68 \times 10^{-3}$ | $5.91 \times 10^{-3}$ |

### 3.4.2. Time-Averaging Uncertainty

Temporal fluctuations in turbulent flows are typically not represented by fluid mechanical calibration factors, meaning that $K_d$ is designed to be independent of time. Nonetheless, the influence of low-frequency turbulent fluctuations is generally not negligible for practical simulation times. The determination of $K_{d.S}$ as an average over a finite time interval $T$ in a transient simulation therefore involves a source of uncertainty, herein referred to as $u_{time}$. Its presence in the simulation setup is connected to the physically correct yet arbitrary state of the initial conditions in the domain and the velocity fluctuations at the inlet boundary at $t = 0$. While $u_{time}$ decreases with an increasing simulation time, it should always be considered in transient simulations.

Local values of $u_{time}$ are estimated using the empirical standard deviation $s(K_{d.S})$ from a total of 10 individual simulation runs with different starting solutions. This approach is similar to a repeatability test in an experimental uncertainty analysis. Analogous to the numerical uncertainties, a representation for the entire domain is obtained by calculating the arithmetic mean. For $T = 120\,\text{s}$, as utilized in the setup for the SBES-SST model, $u_{time}$ is determined as $1.32 \times 10^{-3}$. Other values corresponding to 60 s, 180 s and 240 s are depicted in Figure 7. It is found that the time-dependent development of the uncertainty contribution can be approximated with $u_{time} = a/\sqrt{T \cdot 1\,\text{s}^{-1}}$, where $a = 1.46 \times 10^{-2}$ was determined by a least-squares fit.

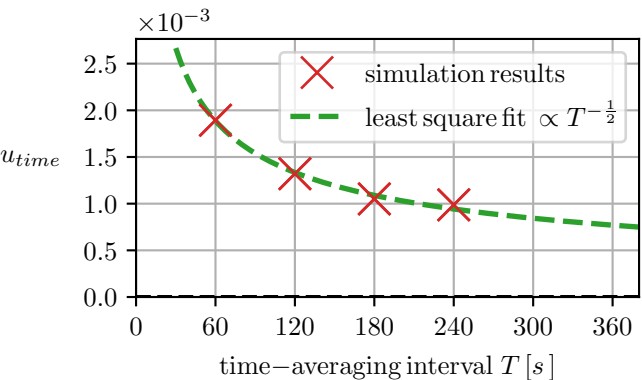

**Figure 7.** Uncertainty contribution due to the finite length of the time-averaging interval.

### 3.4.3. Calculation Uncertainty

It is assumed that the simulation parameters addressed in the verification studies were independent. Thus, the components can be combined in the calculation uncertainty

$$u^2_{cal}(K_{d.S}) = u^2_{disc} + u^2_{wall} + u^2_{iter} + u^2_{time}.$$  (10)

In the case of the steady-state simulations, only $u_{disc}$ and $u_{wall}$ were included in $u_{cal}(K_{d.S})$. Since a confidence interval of 95.45% was targeted, the expanded calculation uncertainty $U_{cal}(K_{d.S}) = k \cdot u_{cal}(K_{d.S})$ with $k = 2$ was introduced. Values of $u_{cal}(K_{d.S})$ and $U_{cal}(K_{d.S})$ calculated for the SBES-SST and Spalart–Allmaras turbulence models are included in Table 3 and Figure 8. It was found that $u_{disc}$ determined for the RANS model exceeded that of the hybrid model by a factor of $\approx 1.7$, whereas for $u_{wall}$, it was the other way around. In combination with the additional components evaluated for the SBES-SST simulation, $u_{cal}(K_{d.S})$ was nearly identical for both models. For a more detailed analysis regarding the impacts of different mesh sizes, time step sizes, and other parameters on the simulation results of double bends, see Straka [17]. Regarding the progression of $u_{time}$ in Figure 7, increasing $T$ from 120 s to 180 s reduces the uncertainty by $\approx 20\%$. This decreases $u_{cal}(K_{d.S})$ in the SBES-SST simulation by only $\approx 2.5\%$ and, thus, does not justify the additional computing time.

In comparison to the fluctuation range of $K_{d.S}$ in Figure 5, $U_{cal}(K_{d.S})$ was smaller by a factor of $\approx 20$. This can be regarded as an indication of a sufficiently high quality of the simulation setup. Nonetheless, the inability to eliminate numerical errors means that the validation experiment following the verification studies not only evaluates modeling errors in particular, but the errors arising from the simulation as a whole.

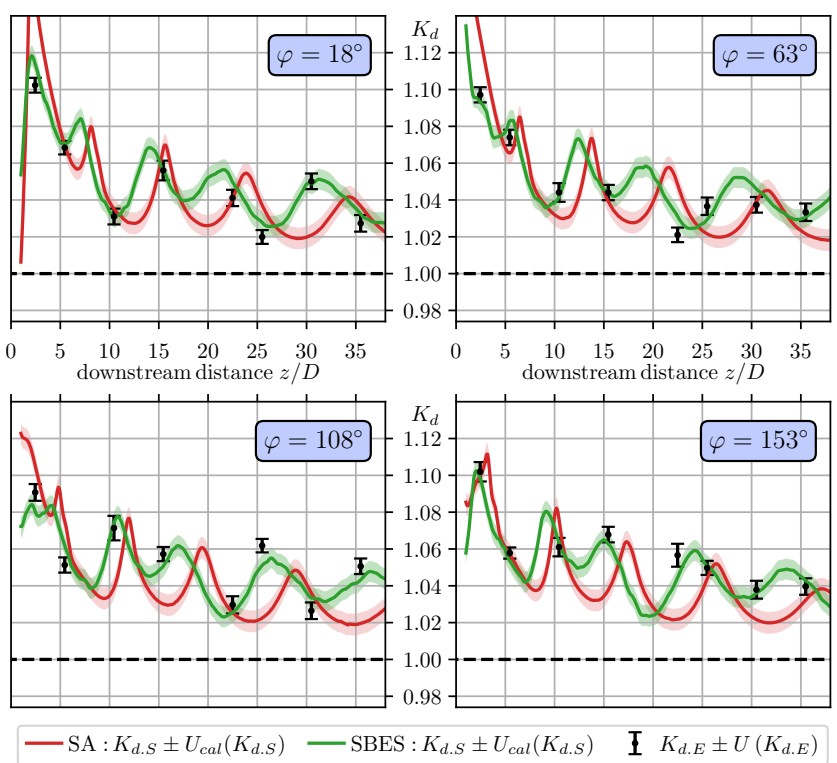

**Figure 8.** Comparison between $K_{d.E} \pm U(K_{d.E})$ and $K_{d.S}$, including the expanded calculation uncertainty $U_{cal}(K_{d.S})$ determined for the hybrid SBES-SST and Spalart–Allmaras (SA) RANS model. In contrast to $U(K_{d.S})$, $U_{cal}(K_{d.S})$ does not cover the modeling uncertainty (compare Figure 9). $U(K_{d.E})$ and $U_{cal}(K_{d.S})$ are represented by error bars and bands, respectively.

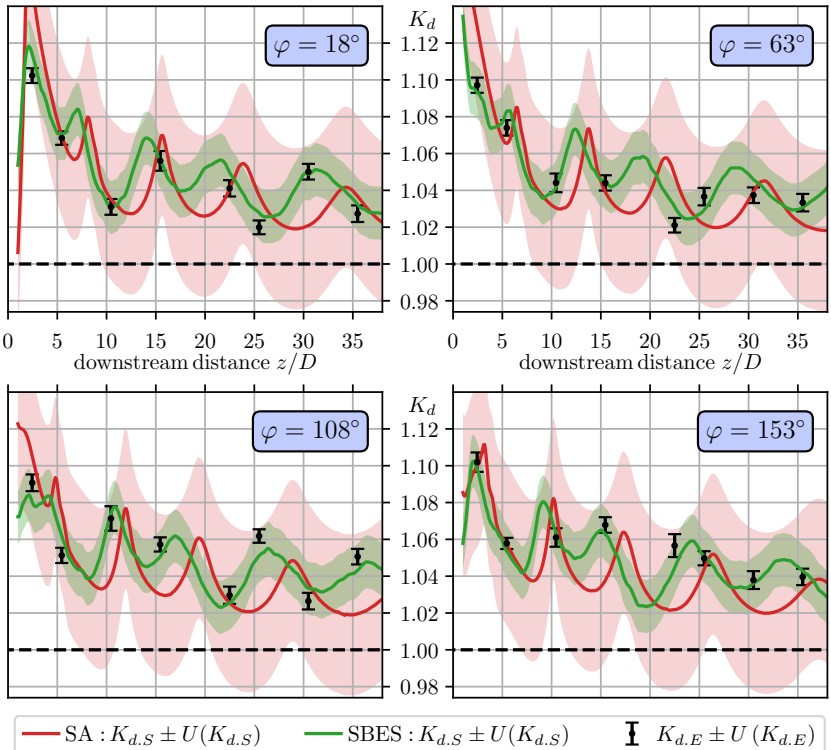

**Figure 9.** Comparison between $K_{d.E} \pm U(K_{d.E})$ and $K_{d.S}$, including the expanded simulation uncertainty $U(K_{d.S})$ determined for the hybrid SBES-SST and Spalart–Allmaras (SA) RANS model. $U(K_{d.E})$ and $U(K_{d.S})$ are represented by error bars and bands, respectively.

### 3.5. Comparison with Experimental Data

A comparison between $K_{d.E}$ and $K_{d.S}$ over the downstream distance is presented in Figure 8. As observed in Figure 5, there is a significant phase shift between $K_{d.S}$ as predicted by the two turbulence models, visible at all angular orientations. In general, the predictions obtained from the simulation with the SBES-SST model are in better agreement with the measurement data. Nonetheless, $K_{d.E} \pm U(K_{d.E})$, in part, lie on the edges or even fall outside the specified uncertainty band of $K_{d.S}$, even though $U_{cal}(K_{d.S})$ describes a confidence interval of 95.45%. This implies that the predominant components in the overall uncertainty budget (especially in the RANS simulation) must stem from modeling errors, most evidently simplifications of the turbulence model.

### 4. Simulation Uncertainty

This section contains the novel approach to assess the expanded simulation uncertainty $U(K_{d.S})$ in analogy to the evaluation of measurement uncertainty according to the GUM [1]. Since the result of a virtual measurement includes an approximation of $K_d$ in the entire computational domain, $U(K_{d.S})$ is supposed to be valid for all angular orientations and downstream positions. Its derivation is based on the magnitude and distribution of simulation errors $\delta(K_{d.S})$ that can be calculated at the measurement positions. Given that the numerical errors could not be corrected during verification in Section 3, $\delta(K_{d.S})$ contains all sources of errors arising from the simulation as a whole. Conditions for a correction of systematic errors and spatial patterns in the error distribution are examined. The individual concepts are illustrated for the SBES-SST model, while the results obtained for the other turbulence models are discussed in Section 4.4.

### 4.1. Simulation Error

A simulation error associated with a particular variable of interest is defined as the difference between the simulation result at a specific location in the computational domain and the true value. This relation also applies to measurement errors. Thus, the errors in the simulation $\delta(K_{d.S})$ and experimental data $\delta(K_{d.E})$, with regard to the determination of the calibration factor $K_d$, are given by

$$\delta(K_{d.S}) = K_{d.S} - K_d \,, \tag{11a}$$

$$\delta(K_{d.E}) = K_{d.E} - K_d \,. \tag{11b}$$

Equation (11a,b) cannot be solved individually, as the true value of $K_d$ is unknown. Yet by inserting Equation (11a) into (11b), $\delta(K_{d.S})$ can be written as

$$\delta(K_{d.S}) = K_{d.S} - K_{d.E} + \delta(K_{d.E}) \,. \tag{12}$$

In this form, $\delta(K_{d.S})$ can be determined at the measurement positions with the restriction that $\delta(K_{d.E})$ must be estimated. All values of $\delta(K_{d.S})$ calculated from the results obtained with the SBES-SST model are depicted in Figure 10. In the figure, the errors in the experimental data $\delta(K_{d.E})$ are represented by the combined measurement uncertainty $u_c(K_{d.E})$. Excluding the experimental errors, the individual values of $\delta(K_{d.S})$ ranged between $-1.30 \times 10^{-2}$ and $1.05 \times 10^{-2}$.

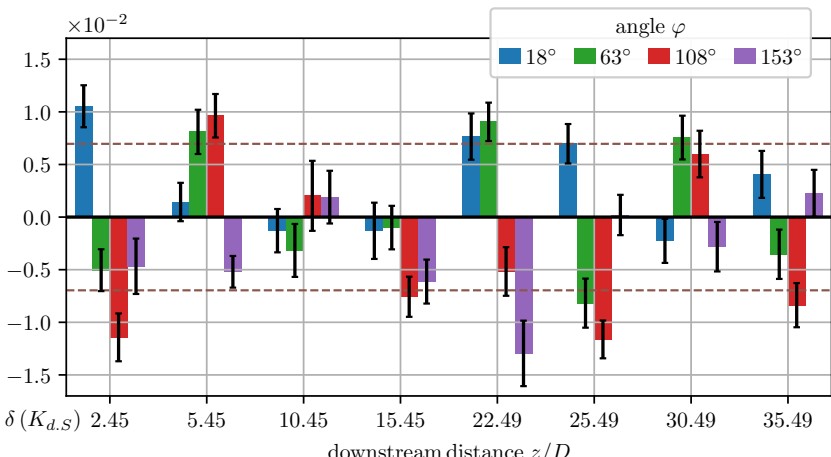

**Figure 10.** Simulation error $\delta(K_{d.S})$ of the SBES-SST model acc. Equation (12). Experimental errors $\delta(K_{d.E})$ are represented by the combined measurement uncertainty $u_c(K_{d.E})$ with black error bars. The brown dashed lines indicate the standard simulation uncertainty $u(K_{d.S})$ acc. Equation (16).

### 4.2. Correction of Systematic Errors

Adapting the simulation results by eliminating systematic errors can be useful for reducing the overall simulation uncertainty. The most basic method is to remove the systematic share by subtracting the arithmetic mean of the simulation errors, such that

$$K_{d.S, corr} = K_{d.S} - \overline{\delta}(K_{d.S}) \tag{13}$$

represents a corrected version of $K_{d.S}$. This adjustment is only adequate if $\overline{\delta}(K_{d.S})$ is significant. We suggest establishing significance under the condition that the absolute mean is greater than its standard error, i.e.,

$$|\overline{\delta}(K_{d.S})| > s(\delta(K_{d.S}))/\sqrt{N}, \tag{14}$$

where $s$ is the empirical standard deviation and $N = 32$ is the number of measurement positions and, thus, the individual values of $\delta(K_{d.S})$. In case of the simulation results obtained with the SBES-SST model, $\overline{\delta} = -7.63 \times 10^{-4}$ and $s = 6.91 \times 10^{-3}$. Thus, $|\overline{\delta}|$ is smaller than its standard error $s/\sqrt{32} = 1.22 \times 10^{-3}$. Consequently, a correction according to Equation (13) is not pursued. As for the other turbulence models investigated in this study, the condition set out in Equation (14) is fulfilled in the case of the SBES-realizable-$k$-$\epsilon$, Spalart–Allmaras, and linear pressure–strain model (see Table 4).

**Table 4.** Simulation uncertainty results of all turbulence models.

| Turbulence Model | $\overline{\delta}(K_{d.S})$ Significant? | $u(K_{d.S})$ | Coverage Factor $k$ | $U(K_{d.S})$ |
|---|---|---|---|---|
| SBES-SST [30] | no | $6.96 \times 10^{-3}$ | 2.07 | $1.44 \times 10^{-2}$ |
| SBES-realizable $k$-$\epsilon$ [30] | yes | $1.29 \times 10^{-2}$ | 2.08 | $2.67 \times 10^{-2}$ |
| DES-realizable $k$-$\epsilon$ [31] | no | $1.42 \times 10^{-2}$ | 2.08 | $2.96 \times 10^{-2}$ |
| DES-SST [31] | no | $1.63 \times 10^{-2}$ | 2.08 | $3.39 \times 10^{-2}$ |
| Spalart-Allmaras [27] | yes | $2.10 \times 10^{-2}$ | 2.08 | $4.38 \times 10^{-2}$ |
| SST $k$-$\omega$ [26] | no | $2.35 \times 10^{-2}$ | 2.08 | $4.89 \times 10^{-2}$ |
| Standard $k$-$\omega$ [25] | no | $2.73 \times 10^{-2}$ | 2.08 | $5.69 \times 10^{-2}$ |
| Linear pressure–strain [28] | yes | $2.78 \times 10^{-2}$ | 2.08 | $5.80 \times 10^{-2}$ |
| Standard $k$-$\epsilon$ [23] | no | $3.06 \times 10^{-2}$ | 2.08 | $6.38 \times 10^{-2}$ |
| Realizable $k$-$\epsilon$ [24] | no | $3.14 \times 10^{-2}$ | 2.08 | $6.55 \times 10^{-2}$ |
| Stress-$\omega$ [25] | no | $3.82 \times 10^{-2}$ | 2.08 | $7.95 \times 10^{-2}$ |

### 4.3. Uncertainty Quantification

Following the specification of the simulation errors, the objective is to provide an expanded uncertainty $U(K_{d.S})$ associated with the prediction of $K_{d.S}$ at arbitrary locations downstream of the double bend, such that the interval $\pm U(K_{d.S})$ contains $\delta(K_{d.S})$ with a probability of 95.45%. Since evaluating $U(K_{d.S})$ is based on the magnitude and spatial arrangement of the simulation errors, the spatial autocorrelation was examined at first.

#### 4.3.1. Spatial Autocorrelation

Clear positive correlations between $\delta(K_{d.S})$ and the downstream distance or angular orientation towards the double bend can be used to attribute corresponding dependencies to the uncertainty. The spatial arrangement of the simulation errors in Figure 10, however, does not seem to follow any discernible pattern within the measurement range. For this reason, we assume that the simulation errors determined for the SBES-SST results can be considered as white noise, meaning that their magnitudes and signs are spatially independent. This hypothesis is confirmed through an evaluation of spatial autocorrelation using Moran's $I$ [35], see Appendix B. Due to the absence of patterns in the spatial distribution, the values of $\delta(K_{d.S})$ at the measurement positions cannot be used to interpolate or extrapolate errors throughout the computational domain. Consequently, the simulation uncertainty is specified to be independent of the downstream distance and angular orientation towards the double bend.

#### 4.3.2. Standard Uncertainty

Standard uncertainty is defined as the estimated standard deviation, which in this case is equivalent to the sample standard deviation $s$. If the systematic share of the simulation errors is eliminated according to Equation (13), the relative standard uncertainty representing the corrected simulation results is therefore given by

$$u(K_{d.S,corr}) = s\big(\delta(K_{d.S,corr})\big). \tag{15}$$

If, on the other hand, $\overline{\delta}(K_{d.S})$ is not significant under the condition given in Equation (13), an equivalent standard uncertainty associated with $K_{d.S}$ can be calculated according to

$$u^2(K_{d.S}) = \overline{\delta}^2(K_{d.S}) + s^2\big(\delta(K_{d.S})\big). \tag{16}$$

Equations (15) and (16) can be solved by means of a Monte Carlo simulation, where $\delta(K_{d.E})$ is randomly sampled from a normal distribution with an expected value of $\mu = 0$ and a standard deviation that corresponds to the combined measurement uncertainty $u_c(K_{d.E})$. As for the (uncorrected) simulation results obtained with the SBES-SST model, solving Equation (16) using a Monte Carlo simulation and a total of $2 \times 10^5$ trails yields $u(K_{d.S}) = 6.96 \times 10^{-3}$. In case of the other turbulence models, the standard simulation uncertainty takes values up to $3.82 \times 10^{-2}$, exceeding SBES-SST by a factor of $\approx 5.5$ (see Table 4). Compared to the hybrid models, the uncertainties calculated for the RANS models are consistently higher.

Since $u(K_{d.S})$ represents a combination of many different numerical and modeling errors, it can be assumed to be approximately normally distributed. If a quantity is normally distributed with the expected value $\mu$ and standard deviation $\sigma$, the interval $[\mu - \sigma, \mu + \sigma]$ contains $\approx 68.3\%$ of the distribution. In case of the SBES-SST results, $\pm u(K_{d.S})$ encompasses 21 out of 32 values of $\delta(K_{d.S})$, which corresponds to $\approx 65.6\%$.

#### 4.3.3. Expanded Simulation Uncertainty

Stating the expanded uncertainty that provides the desired level of confidence of 95.45% requires the specification of a corresponding coverage factor $k$. Thus, $k$, being the most practical option, is estimated by applying the methodology described in the GUM [1], which is based on the usage of $t$-distributions and degrees of freedom ($\text{dof}_{eff}$).

It is found that the solution to Equation (16) obtained by means of a Monte Carlo simulation can be approximated with an accuracy of $\approx \pm 1.5\%$ by applying the propagation of uncertainty according to

$$u^2(K_{d.S}) \approx \widetilde{u}^2(K_{d.S}) + \overline{u}_c^2(K_{d.E}). \tag{17}$$

In the equation, $\overline{u}_c(K_{d.E})$ is the arithmetic mean of $u_c(K_{d.E})$ at the $N = 32$ measurement positions. Furthermore, $\widetilde{u}(K_{d.S})$ is the solution to Equation (16) under the assumption that all experimental errors are zero. This approximation can also be applied to $u(K_{d.S,corr})$. Considering $u(K_{d.S})$ as a combined uncertainty allows the determination of $\mathrm{dof}_{eff}$ using the individual degrees of freedom ($\mathrm{dof}_i$) and the Welch—Satterthwaite equation (Equation (6)). Regarding $\widetilde{u}(K_{d.S})$, $\mathrm{dof}_i$ can be connected to the number of measurements and thus, is determined as $\mathrm{dof}_i = N - 1 = 31$. As for $\overline{u}_c(K_{d.E})$, $\mathrm{dof}_i$ is determined as the arithmetic mean of $\mathrm{dof}_{eff}$ at the 32 measurement positions, which gives $\mathrm{dof}_i = 613$. In case of the simulation results obtained with the SBES-SST model, Equation (6) yields $\mathrm{dof}_{eff} = 38$. The corresponding coverage factor for a confidence interval of 95.45% using the inverse function of the Student's $t$-distribution is calculated as $k = 2.07$. Finally, the expanded simulation uncertainty can be stated as $U(K_{d.S}) = k \cdot u(K_{d.S}) = 1.44 \times 10^{-2}$. Table 4 lists the coverage factors and expanded uncertainties obtained with the other turbulence models.

*4.4. Comparison of Results for Different Turbulence Models*

Table 4 includes the simulation uncertainties obtained with all turbulence models investigated in this study. Furthermore, an updated visualization of the comparison between $K_{d.E}$ and $K_{d.S}$ using $U(K_{d.S})$ instead of $U_{cal}(K_{d.S})$ is presented in Figure 9.

In general, better results were gained with the hybrid RANS-LES turbulence models, as the uncertainties calculated for the RANS models were consistently higher. Among the hybrids, the SBES-SST model exhibited the lowest simulation uncertainty of $U(K_{d.S}) = 1.44 \times 10^{-2}$. As for RANS, the lowest uncertainty was obtained with the one-equation Spalart–Allmaras model in the amount of $U(K_{d.S}) = 4.38 \times 10^{-2}$, surpassing the two-equation RANS and seven-equation Reynolds stress models. Figure 9 reveals that starting at a downstream distance of $\approx 10\,D$, the uncertainty of the RANS model was nearly as large as the measurement error caused by the disturbance. Thus, while using the RANS model for the near-field range may be beneficial, its predictions in the far-field range are not meaningful. In contrast, the uncertainty band of the SBES-SST model still allows for a clear distinction between adjacent installation positions. It can be concluded that the higher uncertainties in case of RANS are connected to the phase shift between the axial velocity profiles illustrated in Figure 6a.

In the transient and steady-state simulations, $U(K_{d.S})$ exceeded $U_{cal}(K_{d.S})$ by a factor of $\approx 4.0$ and 7.4, respectively. Similarly, if $U(K_{d.S})$ is considered as a combined uncertainty, the associated proportion of variance of $U_{cal}$ amounts to only 12.7% in case of SBES-SST and 2.0% in the case of the Spalart–Allmaras model. This clearly suggests that the modeling errors are the dominating contributions to the overall uncertainty.

## 5. Discussion

In this paper, a framework for assessing the simulation uncertainty $U(K_{d.S})$ of a virtual meter realized by means of CFD was proposed. It was conducted on the example of determining fluid mechanical calibration factors for an ultrasonic clamp-on flow meter installed downstream of a double bend out-of-plane. In conformity with the expression of measurement uncertainty according to the GUM [1], $U(K_{d.S})$ was set as an expanded uncertainty providing a confidence interval of 95.45%. Given that the outcome of the virtual flow measurement is not an individual value but an array of equally important values, $U(K_{d.S})$ was designed to be valid for the approximation of $K_d$ at arbitrary installation positions downstream of the disturbance. In accordance with its intended use, the virtual measurement therefore allowed for a continuous determination of calibration factors applicable to the relevant mounting positions of the real meter.

$U(K_{d.S})$ was derived on the basis of simulation errors, which were in turn quantified as functions of the experimentally determined calibration factors $K_{d.E}$, including their corresponding uncertainties $U(K_{d.E})$. Since, as a result, $U(K_{d.S})$ cannot be smaller than $U(K_{d.E})$, the considerable effort put into the determination of $K_{d.E}$ and the quantification of $U(K_{d.E})$ can be justified. Similarly, the virtual meter complies with the requirements established for the *digital metrological twin* as defined by Eichstädt et al. [4], in that $K_{d.S}$ is validated by traceable measurements and $U(K_{d.E})$ calculated according to valid standards. The large number of measurements covering the interesting range of installation positions for the meter is of course time- and cost-intensive and could thus be regarded as a disadvantage of the methodology. However, due to the simplifications and assumptions that are still required for modeling turbulent flows within a feasible time frame, it might be the only suitable way to ensure confidence in a CFD-based virtual meter that fulfills metrological standards.

In contrast to the standard verification and validation approach, where the numerical and modeling uncertainties attributed to a CFD simulation are processed one after the other, $U(K_{d.S})$ describes the uncertainty of the virtual measurement as a whole. On the one hand, evaluating the entire simulation setup in the validation was simply due to the fact that during verification, a correction of the numerical errors by means of the Richardson extrapolation failed. On the other hand, this supports studies (such as Hosder et al. [15]) suggesting that numerical and modeling errors cannot always be separated due to potential interactions between simulation parameters. This applies in particular to transient simulations, where, e.g., a refinement of the mesh results in a resolution of smaller eddy scales and a reduction of the modeled turbulent viscosity. In that regard, achieving grid independence is only possible if the requirements for a direct numerical simulation are fulfilled. Nonetheless, verification studies as conducted in this paper represent a means to detect crucial parameters and minimize sources of errors in the simulation setup. As an alternative to the Richardson extrapolation, numerical errors were incorporated in the expanded calculation uncertainty $U_{cal}(K_{d.S})$ and compared to $U(K_{d.S})$. It was concluded that the modeling errors must be the dominating contributions to the overall uncertainty, especially in the RANS simulation.

Different modeling approaches and commonly used turbulence models were used for the simulation setup of the virtual flow meter. Based on $U(K_{d.S})$, the comparison demonstrated clear superiority of the hybrid RANS-LES turbulence models over the industry standard RANS. Among the hybrids, the best results were achieved with the SBES-SST model. In practical applications, however, the accuracy of a model has to be balanced against the available computing powers. While $U(K_{d.S})$ calculated for the Spalart–Allmaras RANS model exceeded SBES-SST by a factor of $\approx 3$, the computation time required for the transient simulation of the double bend was roughly 100 times higher. This can become impractical, especially if parameter studies with a great number of geometrical variations or Reynolds numbers are involved. On the other hand, it was demonstrated that a virtual measurement carried out with a RANS model is meaningless if its associated uncertainty is in the order of the meter error it is intended to compensate. Despite its comparatively high computing times, a clear recommendation can therefore be made for the use of hybrid RANS-LES turbulence models. As for RANS, the deficiency was related to an inferior prediction regarding the phase position of the axial velocity profiles. Since this is a flow phenomenon typically attributed to double bend out-of-plane configurations, the trade-off between accuracy and computational costs has to be re-evaluated for different flow disturbances.

While the simulation uncertainty calculated in this study may be an appropriate estimation for similar geometries, it is at this point limited to the double bend out-of-plane with a curvature radius of $R_c/D = 1.425$. To accomplish a transferability to other double bend configurations, we suggest further validation studies in the relevant range of curvature radii and distances between the bends. In that case, a small number of measurement positions may be sufficient, provided a similar simulation setup is used. The approach for

quantifying simulation uncertainties as presented here can further be extended to different flow disturbances or other CFD-based virtual flow meter designs. If this incorporates additional dominant flow phenomena, such as flow separation, we recommend a sufficiently large number of experimental data for the validation and a comparison between different turbulence models. Apart from flow applications, the proposed method could further be applied to other fields in metrology, where the evaluation of the overall uncertainty describing a virtual measurement is linked to a validation experiment.

**Author Contributions:** Conceptualization, M.S., A.W. and S.S.; methodology, M.S., A.W. and S.S.; software, M.S.; validation, M.S., A.W., C.K., C.H. and S.S.; formal analysis, M.S.; investigation, M.S. and C.K.; data curation, M.S. and C.K.; writing—original draft preparation, M.S., A.W. and S.S.; writing—review and editing, C.K. and C.H.; visualization, M.S. All authors have read and agreed to the published version of the manuscript.

**Funding:** This research received no external funding.

**Data Availability Statement:** The data presented in this study are available on reasonable request from the corresponding author.

**Acknowledgments:** We would like to thank Clemens Elster and Gerd Wübbeler from the Physikalisch-Technische Bundesanstalt (PTB) for their advice concerning uncertainty quantification and statistical analysis. Furthermore, we thank Bernhard Funck from FLEXIM Flexible Industriemesstechnik GmbH for his assistance in evaluating the experimental uncertainty.

**Conflicts of Interest:** The authors declare no conflict of interest.

## Appendix A. Detailed Description of the Measurement Uncertainty

*Appendix A.1. Uncertainty Estimated for the Determination of Flow Rates*

In order to estimate the uncertainty for the determination of the flow rate $\dot{V}$, all influencing factors in Equation (2) must be taken into account. These parameters can be divided into contributions from the installation ($A$, $K_p$), the meter ($K_g$, $t_0$, $t_{tr}$) and the transit time difference ($\Delta t$). A detailed description can be found in ISO 12242 [36]. At low velocities, the uncertainty associated with the measured transit time difference $u(\Delta t)$ becomes an important factor. It can be divided into a random and systematic part representing the zero-flow offset. According to the manufacturer's specifications, $u(\Delta t)$ is inversely proportional to the transducer frequency $f_0$ and can be approximated using a proportionality constant of $\approx 3 \times 10^{-4}$. This estimate was obtained from transmitter calibrations with synthetic signals. FLEXIM FLUXUS F721 operate at a transducer frequency of $f_0 = 2\,\text{MHz}$ and thus, $u(\Delta t) = 3 \times 10^{-4}/f_0 = 1.5 \times 10^{-10}\,\text{s}$. Moreover, a volumetric velocity of $0.4\,\text{m}\,\text{s}^{-1}$ corresponds to a time difference of $5.6 \times 10^{-8}\,\text{s}$. As a result, the relative uncertainty with regard to $\dot{V}$ is $u_r(\dot{V}_{\Delta t}) = 1.5 \times 10^{-10}\,\text{s}/5.6 \times 10^{-8}\,\text{s} = 2.68 \times 10^{-3}$. In combination with the remaining contributions, the expanded relative uncertainty with respect to the determination of flow rates under fully developed flow conditions at the current setup is roughly estimated at $U_r(\dot{V}) \approx 1\%\,(k=2)$.

*Appendix A.2. Contributions to the Combined Measurement Uncertainty*

For the uncertainty evaluation associated with the experimental determination of $K_d$, the terms on the right side of Equation (5) must be further divided into manageable components. Similar to flow rate measurements, an independent uncertainty evaluation for either $K_u$ or $K_p$ would have to cover all the input quantities for $\dot{V}$ included in Equation (2). Yet when both measurements are combined, it is a good approximation to assume that the parameters associated with the installation and the meter itself, i.e., all except the random part of $\Delta t$, remain constant during the determination of $K_u$ and $K_p$. This means that the correlation coefficients associated with the uncertainties of each pair of parameters, e.g., $u(A_1)$ and $u(A_2)$, equal one and, thus, their respective variances add up to zero. While in theory, this eliminates the influence of $A$, $t_0$, $t_{tr}$, $K_g$, and the systematic share of $\Delta t$, it cannot be entirely ruled out that in reality, the parameters do not remain unchanged

throughout the entire measurement campaign. That also applies to the influence of temperature fluctuations at the test rig. As the temporal stability cannot be readily quantified, this potential source of uncertainty is dealt with in a reproducibility test. This is realized through comparative measurements at the beginning and end of the measurement series and results in the relative uncertainty contributions $u_r(K_{u,rep})$ and $u_r(K_{p,rep})$. Temperature effects are derived from an analysis of previous measurements, which yield $u_r(K_{u,T})$ and $u_r(K_{p,T})$. As for the uncertainty associated with the measured transit time difference $\Delta t$, its random and systematic shares can hardly be separated. For this reason, the uncorrelated random share of $u(\Delta t)$ is estimated in a repeatability test, calculated with the experimental standard deviation of the mean with respect to the three repeated measurements of $K_u$ and $K_p$, respectively. The resulting relative uncertainty contributions are denoted as $u_r(\overline{K_u})$ and $u_r(\overline{K_p})$. In case of the disturbed measurements, we further evaluate uncertainties regarding the angular alignment ($u_r(K_{p,\varphi})$) and downstream distance ($u_r(K_{p,z})$) between the meters and the double bend. Measuring resolutions and environmental conditions are assumed to be negligible. As for the uncertainties associated with $\dot{V}_{ref}$ in Equation (3), the reference meter is not affected by the position of the clamp-on meters and thus, $u_r(\dot{V}_{ref,u}) = u_r(\dot{V}_{ref,p}) = u_r(\dot{V}_{ref})$. In sum, the combined relative uncertainty $u_{c,r}(K_{d.E})$ formerly derived from Equation (4) can be written as

$$
\begin{aligned}
u_{c,r}^2(K_{d.E}) = {} & u_r^2(\overline{K_u}) + u_r^2(K_{u,rep}) + u_r^2(K_{u,T}) + 2 \cdot u_r^2(\dot{V}_{ref}) \\
& u_r^2(\overline{K_p}) + u_r^2(K_{p,rep}) + u_r^2(K_{p,T}) + u_r^2(K_{p,\varphi}) + u_r^2(K_{p,z}).
\end{aligned}
\tag{A1}
$$

A representative calculation example for $z/D = 35.5$ and $\varphi = 18°$ is presented in Table A1. The calculation of the individual uncertainty components in Equation (A1) is described in the following.

Repeatability: At each measurement position, $K_p$ and $K_u$ are calculated as the arithmetic mean from $N = 3$ repeated measurements. The associated uncertainties $u_r(\overline{K_u})$ and $u_r(\overline{K_p})$ are estimated using the experimental standard deviation of the mean $s$, which are normalized by the mean to obtain a relative contribution. This yields

$$
u_r(\overline{K_u}) = \frac{s(\overline{K_u})}{K_u} = \frac{s(K_u)}{\sqrt{N} \cdot K_u} \quad \text{and}
\tag{A2a}
$$

$$
u_r(\overline{K_p}) = \frac{s(\overline{K_p})}{K_p} = \frac{s(K_p)}{\sqrt{N} \cdot K_p}.
\tag{A2b}
$$

Throughout the measurement positions, $u_r(\overline{K_u})$ varies from $2.79 \times 10^{-5}$ to $7.90 \times 10^{-4}$, whereas $u_r(\overline{K_p})$ ranges from $7.42 \times 10^{-5}$ to $1.06 \times 10^{-3}$. Their corresponding degrees of freedom are determined as $\mathrm{dof}_i = N - 1 = 2$, respectively.

Reproducibility: The measurement series under fully developed flow conditions was carried out prior to the investigations with the double bend (at time $t = t_1$) and repeated afterward ($t = t_2$). Reproducibility can therefore be tested with regard to the temporal aspect of the measurement campaign, considering both the reconfiguration of the experimental setup and the mid-term stability of the sensors. From each pair of calibration factors measured with the eight meters, $u_r(K_{u,rep})$ is estimated as

$$
u_r(K_{u,rep}) = \frac{|K_u(t_1) - K_u(t_2)|}{\sqrt{12} \cdot \overline{K_u(t_{1,2})}},
\tag{A3}
$$

assuming a rectangular distribution. In the equation, $\overline{K_u(t_{1,2})}$ represents the arithmetic mean of $K_u(t_1)$ and $K_u(t_2)$, which is used to create a relative measure. Among the eight meters, $u_r(K_{u,rep})$ varies from $5.78 \times 10^{-5}$ to $1.19 \times 10^{-3}$. Even though reproducibility was tested under fully developed conditions only, the measurements downstream of the double bend were also affected equally. Thus, the additional term $u_r(K_{p,rep}) = u_r(K_{u,rep})$

is included in Equation (A1). Given that they were obtained from a Type B evaluation, the associated degrees of freedom are determined as $\text{dof}_i \to \infty$.

Temperature effects: In the control mode, the water temperature of the test rig oscillates periodically around the set point with maximum amplitudes of $\hat{T} = 0.4\,\text{K}$ (Straka [17]). The influence on the ultrasonic clamp-ons is estimated from previous measurements conducted with 6 identical meters at fully developed flow conditions. The campaign covered a temperature range from 20 °C to 60 °C taken in steps of 10 K each. From the temperature curves, an averaged relative temperature dependence of $m_T = 6.02 \times 10^{-4}\,\text{K}^{-1}$ can be identified with regard to $K_u$. Given that the standard deviation of a sinusoidal temperature distribution equals the effective value $T_{eff} = \hat{T}/\sqrt{2}$, the corresponding relative standard uncertainty is calculated as

$$u_r(K_{u,T}) = T_{eff} \cdot m_T = 1.70 \times 10^{-4}. \tag{A4}$$

This estimation is independent of the measurement position, yet it affects both $K_u$ and $K_p$. For this reason, $u_r(K_{p,T}) = u_r(K_{u,T})$ is included in Equation (A1). From 6 meters and 5 temperature curves, $\text{dof}_i$ are determined as $(6 \cdot 5) - 1 = 29$.

Angular alignment and axial distance: As depicted in Figure 8, $K_{d.E}$ and $K_{d.S}$ show a strong dependency on both the angular alignment and downstream distance towards the double bend. The accuracy at which the clamp-on meters can be installed is therefore a significant source of uncertainty for $K_{d.E}$. With the present setup, standard uncertainties for the angular alignment and downstream distance are estimated at 3° and 5 mm, the latter mainly connect to the manufacturing of the double bend. As a result of welding and the subsequent removal of protruding material between pipe fitting and flange, the beginning of the straight pipeline is not identical throughout the cross-section. Both uncertainty contributions to $K_{d.E}$ are determined separately using the simulation results and Monte Carlo experiments. For that purpose, the discrete values of $K_{d.S}$ are replaced by Akima splines (Akima [37]). In proximity to the measurement positions, the continuous spline functions $K_{d.S}(\varphi)$ and $K_{d.S}(z/D)$ are then sampled at random locations using normal distributions with standard uncertainties of 3° and 5 mm, respectively. At a total of $N = 10{,}000$ trials, this yields probability distributions for $K_{d.S}$. Their standard deviations correspond to the desired uncertainty contributions and are specified as $u_r(K_{p,\varphi})$ and $u_r(K_{p,z})$. Throughout the measurement positions, $u_r(K_{p,\varphi})$ varies from $7.09 \times 10^{-5}$ to $2.28 \times 10^{-3}$, whereas $u_r(K_{p,z})$ ranges from $1.08 \times 10^{-4}$ to $1.12 \times 10^{-3}$. These significant fluctuations are related to the irregular development of the velocity profiles in the circumferential and axial directions. In accordance with the number of samples, $\text{dof}_i$ are determined as $N - 1 = 9999$.

**Table A1.** Measurement uncertainty evaluation for $K_{d.E} = 1.027$ at $z/D = 35.5$ and $\varphi = 18°$.

| Part | Description | Symbol | Relative Standard Uncertainty $u_i$ | Relative Variance $u_i^2$ | Proportion of Variance in [%] | Degrees of Freedom $\text{dof}_i$ |
|---|---|---|---|---|---|---|
| $u_r(K_u)$ | Reproducibility | $u_r(K_{u,rep})$ | $1.03 \times 10^{-3}$ | $1.06 \times 10^{-6}$ | 22.45 | $\infty$ |
| | Reference flow rate | $u_r(\dot{V}_{ref})$ | $7.74 \times 10^{-4}$ | $5.99 \times 10^{-7}$ | 12.67 | 82 |
| | Repeatability | $u_r(\overline{K_u})$ | $4.78 \times 10^{-4}$ | $2.28 \times 10^{-7}$ | 4.83 | 2 |
| | Temperature effects | $u_r(K_{u,T})$ | $1.70 \times 10^{-4}$ | $2.89 \times 10^{-8}$ | 0.61 | 29 |
| $u_r(K_p)$ | Reproducibility | $u_r(K_{u,rep})$ | $1.03 \times 10^{-3}$ | $1.06 \times 10^{-6}$ | 22.45 | $\infty$ |
| | Reference flow rate | $u_r(\dot{V}_{ref})$ | $7.74 \times 10^{-4}$ | $5.99 \times 10^{-7}$ | 12.67 | 82 |
| | Angular alignment | $u_r(K_{p,\varphi})$ | $7.22 \times 10^{-4}$ | $5.21 \times 10^{-7}$ | 11.03 | 9999 |
| | Repeatability | $u_r(\overline{K_p})$ | $7.13 \times 10^{-4}$ | $5.08 \times 10^{-7}$ | 10.76 | 2 |
| | Downstream distance | $u_r(K_{p,z})$ | $3.01 \times 10^{-4}$ | $9.06 \times 10^{-8}$ | 1.92 | 9999 |
| | Temperature effects | $u_r(K_{u,T})$ | $1.70 \times 10^{-4}$ | $2.89 \times 10^{-8}$ | 0.61 | 29 |
| Combined relative variance $u_{c,r}^2(K_{d.E})$ | | | | $4.73 \times 10^{-6}$ | 100.00 | |
| Combined relative standard uncertainty $u_{c,r}(K_{d.E})$ | | | | $2.17 \times 10^{-3}$ | | |
| Effective degrees of freedom $\text{dof}_{eff}$ | | | | 136 | | |
| Coverage factor $k$ for the confidence level of 95.45% | | | | 2.02 | | |
| Expanded relative uncertainty $U_r(K_{d.E})$ | | | | $4.39 \times 10^{-3}$ | | |

## Appendix B. Detection of Spatial Patterns in the Simulation Errors

For the derivation of the simulation uncertainty in Section 4, it is assumed that the spatial arrangement of the simulation errors $\delta(K_{d.S})$ does not exhibit any discernible patterns. This hypothesis was tested through an evaluation of spatial autocorrelation using Moran's $I$ [35] and its associated significance parameters. Moran's $I$ is a commonly used correlation coefficient that can take values between $-1$ and $1$. Under the assumption of no spatial autocorrelation, its expected value is $\mu(I) = -1/(N-1)$, where $N$ is the number of spatial units, which are the measurement positions in our case. It describes the overall tendency of neighboring elements to be either similar (values significantly above $\mu(I)$), dispersed (values significantly below $\mu(I)$), or independent (values $\approx \mu(I)$). The definition of neighboring areas by means of a weight matrix has great influence and must therefore be specified appropriately. The statistical significance of the analysis is established through an evaluation of z-scores and p-values. A Z-score is defined as the difference between Moran's $I$ and $\mu(I)$ expressed as a multiple of the standard deviation associated with a randomly generated reference distribution. The corresponding p-value can be interpreted as the probability that the present pattern is generated randomly.

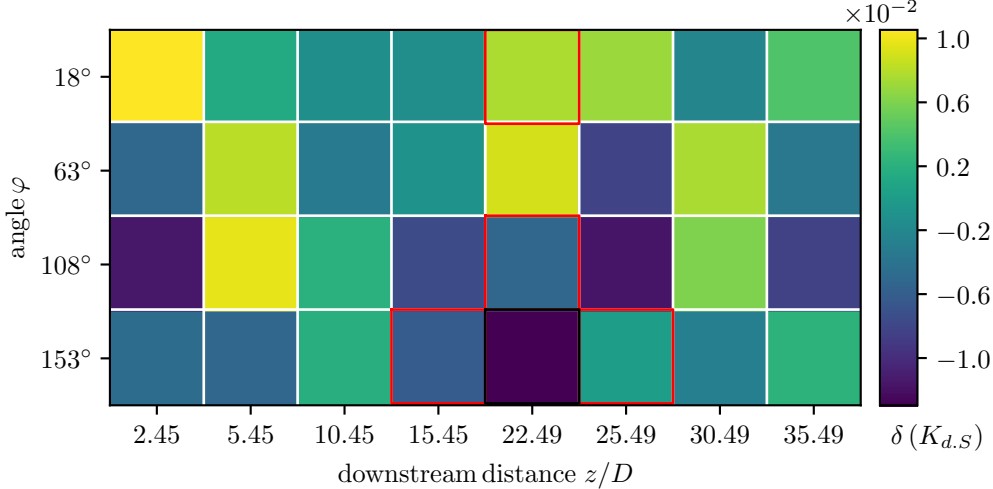

**Figure A1.** Definition of neighboring areas for the simulation errors $\delta(K_{d.S})$ within the evaluation of spatial autocorrelation. The four fields bordered in red indicate the specified neighbors for the field at $\varphi = 153$ and $z/D = 22.5$ bordered in black.

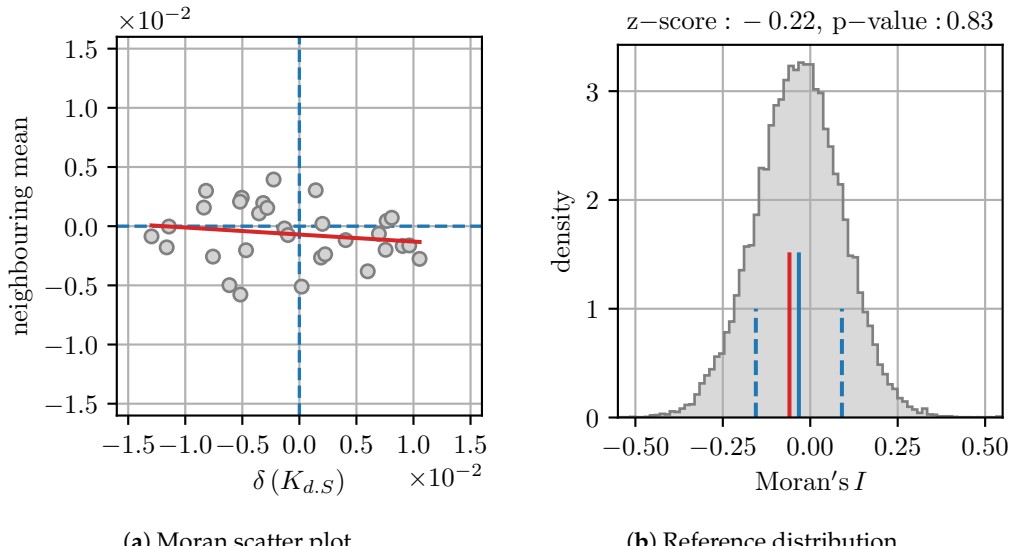

(**a**) Moran scatter plot.

(**b**) Reference distribution

**Figure A2.** Evaluation of spatial autocorrelation using Moran's $I$, which is determined as $-0.06$. The reference distribution has an expected value of $-0.03$ and a standard deviation of $\pm 0.12$.

The derivation of Moran's *I* and associated z-score for the relative simulation errors of the SBES-SST results are illustrated in Figures A1 and A2. As demonstrated in Figure A1, neighboring elements receiving a weight of 1 are defined as the 3–4 adjacent measurement positions in the axial and circumferential direction. The Moran scatter plot depicted in Figure A2a relates each value of $\delta(K_{d.S})$ to the arithmetic mean of its neighboring values. Moran's *I* corresponds to the slope of the least squares regression line between the neighboring means. In case of the SBES-SST results, it has a value of $I = -0.06$, while the expected value $\mu(I)$ is calculated as $-0.03$. Significance parameters are obtained by creating a reference distribution as depicted in Figure A2b. It is generated by randomly shuffling the 32 values of $\delta(K_{d.S})$ in a total of $2 \times 10^5$ trails. In each trail, Moran's *I* is calculated. The expected value of the distribution is $\mu(I)$. Its standard deviation is calculated as $\sigma(I) = \pm 0.12$, resulting in a z-score of 0.22 and *p*-value of 0.83 Thus, the hypothesis of no spatial autocorrelation is confirmed. Similarly, describing the errors as random white noise is a plausible assumption. Moran's *I* and its associated significance parameters for the other turbulence models investigated in this study are listed in Table A2.

**Table A2.** Moran's *I* and its associated significance parameters for all turbulence models.

| Modeling Approach | Turbulence Model | Autocorrelation | | |
| --- | --- | --- | --- | --- |
| | | Moran's *I* | z-Score | *p*-Value |
| Hybrid LES-RANS | SBES-SST [30] | −0.06 | −0.22 | 0.83 |
| | SBES-realizable $k$-$\epsilon$ [30] | −0.09 | −0.43 | 0.66 |
| | DES-realizable $k$-$\epsilon$ [31] | −0.10 | −0.53 | 0.60 |
| | DES-SST [31] | −0.13 | −0.82 | 0.41 |
| RANS | Spalart-Allmaras [27] | 0.11 | 1.21 | 0.23 |
| | SST $k$-$\omega$ [26] | −0.14 | −0.89 | 0.38 |
| | Standard $k$-$\omega$ [25] | −0.14 | −0.86 | 0.39 |
| | Linear pressure–strain [28] | −0.19 | −1.28 | 0.20 |
| | Standard $k$-$\epsilon$ [23] | −0.09 | −0.48 | 0.63 |
| | Realizable $k$-$\epsilon$ [24] | −0.18 | −1.19 | 0.24 |
| | Stress-$\omega$ [25] | −0.22 | −1.57 | 0.12 |

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
