# Peer review of "Simulation Uncertainty for a Virtual Ultrasonic Flow Meter"

_2673-8244, doi:10.3390/metrology2030021_

Round 1
Reviewer 1 Report
The manuscript “Simulation uncertainty for a virtual ultrasonic flow meter” by Straka et al. reports a framework for assessing the overall uncertainty of a virtual flow meter. This work is well written and could be accepted with minor revision. Here are the comments and suggestions:
- Some important results could be added in the abstract.
- Numbers of Tables and Figures should not be the same in the Appendix with that in the main text.
- Citations could be moved to the additional latest column in Tables 4 and S2.
Author Response
Dear reviewer,
thank you very much for your helpful comments and suggestions.
Point 1: In fact, the original abstract does not contain any results or conclusions. We have added them at the end of the abstract.
Point 2: We also noticed this issue. With certain latex editors, the separate numbering in the appendix seems to work properly (e.g., Figure A.1). With other editors, such as in overleaf, it does not. We will consult our editor in charge at MDPI on this matter.
Point 3: We considered doing that, but unfortunately the table would than extend the width of the paper, so we kept it as before.
Kind regards,
Martin Straka and co-authors
Reviewer 2 Report
1. Introduction - calibration factors Kd is understandable only to a small group of specialists. In the Introduction, I propose to explain this concept in more detail so that it would be understandable to a wider group of readers. Because the formula explaining this parameter is presented in as many as 140 lines of the paper.
2. The error of a typical electromagnetic flow meter is approx. 1%, for ultrasonic flow meters it is approx. 2%. Is the accuracy of the reference flow meter too low? Please comment on that. This is important due to equation 3. What is the uncertainty of the determination of this coefficient, which has a direct impact on the uncertainty of Kd.E (eq.5). With the values of this uncertainty of 10-3, it requires a more detailed explanation.
3. The authors conducted a series of numerical tests and presented the simulation results. There is no doubt that the choice of turbulence model will have an impact on the results obtained. The study lacks a comparison of the results obtained from different models and no assessment of the influence of these models on the simulation result.
4. It is worth carrying out a similar discussion for different computational meshes.
Author Response
Dear reviewer,
thank you very much for your helpful comments and suggestions.
Point 1: The concept of using calibration factors was indeed introduced only very briefly. Thank you for pointing that out. We have added some more information regarding the usage as a constant of proportionality, its potential range and implementation in the first paragraph of the introduction.
Point 2: While a typical electromagnetic flow meter may certainly have an uncertainty up to 1\%, the KROHNE WATERFLUX 3070 used at the test rig was calibrated at PTB's gravimetric heat meter facility for the exact same temperature and flow rate we used for the investigations with the ultrasonic clamp-ons (we added this information to Sec. 2.1.2). It was further characterized for the flow conditions at the LULA resulting in an uncertainty for the flow rate determination of 7.74e-4, which is also stated in Sec. 2.1.2. Since the (standard) uncertainty of the clamp-ons regarding a flow rate determination is estimated at 5e-3, the combined uncertainty of Kp (equation 3) would be mainly dominated by the clamp-ons. In other words, the accuracy of the reference meter is almost a magnitude lower and should therefore be sufficient in our opinion. However, an individual determination of Kp is not pursued and may not even be advisable from our point of view. Instead, we combine the measurements under fully developed and disturbed flow conditions, resulting in KdE according to Equation 4. This has the advantage, that most of the input quantities for the clamp-on flow rate can be neglected, because they remain constant in between the measurements for Kp and Ku. As for the original paper, this main advantage of our procedure is hidden and explained only in Annex A. Following your inquiry, a brief summary has been added to Sec. 2.3, complemented with a reference for Annex A. We hope, this resolves your question in a satisfactory manner.
Point 3: We agree with you that the presentation of the simulation results obtained with all turbulence models is by no means complete. This is for one thing due to the length of the paper, that is already very long. On the other hand, including the results of other models would in our opinion not enhance the informational value of the study. For the sake of clarity we choose to cover the results of one hybrid RANS-LES model and one RANS model in detail to explain and discuss the overall approach presented in the paper instead. This includes the verification studies, the development of the velocity profiles and an assessment of the simulation uncertainties. Since the determination of the simulation uncertainty is the main aspect, we further decided to include the results with all turbulence models in Table 4, with a brief description of the results in Sec. 4.4. We hope you'll find this plausible, as a further extension of the results would in our opinion make it more difficult to grasp the essential part of the paper.
Point 4: Again, you are addressing an interesting aspect that could not be treated in full detail due to the length of the paper. As an alternative, we added an additional reference in Sec. 3.4.3 that provides a detailed analysis of different mesh sizes. This also includes attempts to classify the various results in terms of the underlying physics. As for the current paper, the influence of the mesh on Kp is included in the calibration uncertainty and depicted and discussed (briefly) in Fig. 8 and Sec. 3.4.3. Based on the uncertainty bands displayed in Fig. 8, it can be deduced that one would hardly recognise the differences in the velocity profiles or the course of Kp when plotted next to each other. What is most interesting within the scope of this study is that all of the numerical uncertainties dealt with in the verification are small compared to the modeling uncertainties.
Kind regards,
Martin Straka and co-authors
Reviewer 3 Report
This paper developed a method for assessing the uncertainty of a virtual ultrasonic flow meter. The framework proposed by the authors is a good guideline for evaluating the so-called virtual flow meter. Basically,the paper is well written and organized, and the results presented are satisfying. I have no more suggestions than to suggest that the author check some spelling mistakes.
Author Response
Dear reviewer,
thank you very much for your positive feedback. We thoroughly looked for spelling mistakes and corrected a few that we could find.
Kind regards,
Martin Straka
Reviewer 4 Report
The authors present a framework for analyzing simulation uncertainty obtained from CFD. They obtain calibration factors from experimental results and compare those to different models in CFD. The advantage of using simulations is the continuous determination of calibration factors for different mounting positions, as compared to a more costly experimental setup. Overall more uncertainty was observed in RANS models when compared to hybrid RANS-LES models. It comes out as a trade-off between the computational cost to accuracy.
Overall, the work is interesting and pays good attention to detail. The paper is publishable after addressing some minor comments.
Minor Comments:
The data presented in Sections 2.1.1 and 2.1.2 are given in SI units, which is perfectly fine. It would also be helpful to report the Reynolds number based on shear stress (friction Reynolds number) and the velocities and surface roughness in plus units for example, as it gives a better idea of the scaling involved.
In Section 2.2 the path velocity is a function of Kg the path-geometry factor and the ratio of (\Delta t) / (t_{tr}-t_o) in Equation 1. It is not obvious if the units of velocity are consistent. Can this be elaborated on?
In Section 2.5 line 203, the claim of interpolating between the measurement position is not meaningful due to angular dependency. Why not find a non-dimensionalization that collapses them instead? Is that possible? Can the authors comment on that? Since at each angle a similar behavior in trend is observed.
Appendix A1: Line 597 flow velocity (\Delta t). This was defined earlier in Section 2.2 as the transit time difference.
Author Response
Dear reviewer,
thank you very much for your positive feedback and the helpful comments and suggestions.
Point 1: We added the friction velocity and Reynolds number as well as a dimensionless roughness parameter in Sec. 2.1.
Point 2: As a constant of proportionality, Kg has the unit of a velocity. We added this information to Sec. 2.2 following Eq. 1.
Point 3: That is indeed an interesting idea. In fact, the flow profiles downstream the double bend out-of-plane show some systematic behavior (cf.~Fig.~6). Hence, one could try to transform the results in such a way that the calibration factors for the different measurement angles collapse. Such an approach can be realized for CFD results, where a continuous development of the calibration factors is available. Due to the low resolution of the experimental data, this approach cannot be used here, because the oscillations are not represented sufficiently. Thus, with the knowledge gained from the experimental data only, no meaningful transformation can be constructed.
Point 4: Delta t is indeed the transit time difference. This has been corrected in Appendix A1.
King regards,
Martin Straka and co-authors